# Preparation of a Mussel-Inspired Supramolecular Polymer Coating Containing Graphene Oxide on Magnesium Alloys with Anti-Corrosion and Self-Healing Properties

**DOI:** 10.3390/ijms24054981

**Published:** 2023-03-05

**Authors:** Meiling Zhang, Xiaoming Yu, Mengyi Sheng, Hua Chen, Bailin Chen

**Affiliations:** School of Materials Science and Engineering, Changchun University of Technology, Changchun 130012, China

**Keywords:** supramolecular polymer, self-healing, anti-corrosion, graphene oxide, cerium conversion

## Abstract

Herein, we present a mussel-inspired supramolecular polymer coating to improve the an-ti-corrosion and self-healing properties of an AZ31B magnesium alloy. A self-assembled coating of polyethyleneimine (PEI) and polyacrylic acid (PAA) is a supramolecular aggregate that takes advantage of the weak interaction of non-covalent bonds between molecules. The cerium-based conversion layers overcome the corrosion problem between the coating and the substrate. Catechol mimics mussel proteins to form adherent polymer coatings. Chains of PEI and PAA interact electrostatically at high density, forming a dynamic binding that causes strand entanglement, enabling the rapid self-healing properties of a supramolecular polymer. The addition of graphene oxide (GO) as an anti-corrosive filler gives the supramolecular polymer coating a superior barrier and impermeability properties. The results of EIS revealed that a direct coating of PEI and PAA accelerates the corrosion of magnesium alloys; the impedance modulus of a PEI and PAA coating is only 7.4 × 10^3^ Ω·cm^2^, and the corrosion current of a 72 h immersion in a 3.5 wt% NaCl solution is 1.401 × 10^−6^ Ω·cm^2^. The impedance modulus of the addition of a catechol and graphene oxide supramolecular polymer coating is up to 3.4 × 10^4^ Ω·cm^2^, outperforming the substrate by a factor of two. After soaking in a 3.5 wt% NaCl solution for 72 h, the corrosion current is 0.942 × 10^−6^ A/cm^2^, which is superior to other coatings in this work. Furthermore, it was found that 10-micron scratches were completely healed in all coatings within 20 min, in the presence of water. The supramolecular polymer offers a new technique for the prevention of metal corrosion.

## 1. Introduction

Magnesium alloys have played an indispensable role in the shipbuilding, automotive, and aerospace sectors due to their high specific strength, superior malleability, and low density [1,2,3]. However, the standard electrode potential for magnesium is −2.73 V., which makes it more susceptible to dissolution in corrosive chlorine-containing media or humid environments [4,5]. The poor corrosion/wear resistance severely limits their long service life [6]. Metal corrosion has become an impediment to the progress of human society; China’s annual economic losses caused by corrosion account for about 5 % of the GDP. Recently, organic coatings have been found to be the most efficient and cost-effective solutions for enhancing corrosion resistance and extending the service life of magnesium alloys. Traditional coatings [7] act as a passive physical barrier, so they are unable to offer long-term protection because of their low adherence [8]. Currently, the incorporation of two-dimensional (2D) materials into a polymer matrix, such as graphene oxide (GO) [9], hexagonal boron nitride [10], glass sheet [11], layered double hydroxide [8], and others, is a promising technique to improve the corrosion performance of coatings. Researchers are starting to pay more attention to GO due to its strong chemical activity, improved water dispersion, and superior lubricity [12,13,14]. Herein, GO was added to the composite coating as an anti-corrosive filler because of its superior barrier properties and impermeability [15,16].

According to a recent study, graphene-coated metals may corrode even more quickly than bare metals [17], despite numerous reports claiming positive results regarding their corrosion resistance. There are considerable discrepancies in the research results due to the uneven surface coverage and the existence of flaws, such as creases and fissures [18,19]. These flaws are the main cause of poor corrosion resistance in applications that persist for a long time. Research has shown that graphene’s excellent electrical conductivity stimulates the electrochemical corrosion of metals [17]. It has been demonstrated that when metal and graphene are combined, corrosion microcells form in the damaged regions, where the metal acts as an anode to promote electrochemical corrosion. Magnesium alloys may quickly react with graphene oxide and polyelectrolyte solutions during the preparation process, which can improve the capacity for the Mg substrates to dissolve [20]. According to reports by Cao et al. [21], graphene oxide flakes and metal materials like Zn and Cu exhibit a redox reaction and erosion behavior. A cerium-based conversion layer has a high adsorption capacity for Mg alloy substrates [22,23], a denser structure, and less electrical conductivity than the Mg substrate’s native oxide film [24]. Therefore, it can effectively block the interaction of graphene oxide and polymers with Mg alloys, providing a high degree of corrosion protection.

Due to their weak connection, the graphene layers have less adhesion to the metal beneath them. Although some surface modifications and techniques for functionalizing graphene can improve coating adhesion, these techniques may not be sufficient for long-term durability. Dopa is a common chemical modification component that is safe for the environment, which was developed in response to the powerful adhesion of mussels to rocks [25]. Dopa can be deposited on many different types of materials, such as metals [26], biomass [27], plastics [28], and nanoparticles [29]. Numerous catechol functional groups in dopa have been demonstrated to have strong physical adhesion to the substrate surface [30,31]. Therefore, to solve the poor bond between the coating and the surface of the magnesium alloy, here we introduced catechol into organic coatings such that the catechol acts as a binder, which facilitates the preparation of supramolecular anti-corrosion coatings with sustained long-term effects.

The layer-by-layer assembly (LbL) method was used in this research to create a conventional Ce(IV)/PEI-C/PAA-GO supramolecular polymer coating on a Mg alloy substrate. The catechol was grafted onto PEI to maintain the cationic nature while simulating the high catechol content of mussel-sticky proteins. The PAA was cross-linked with GO via hydrogen bonding as the primary corrosion filler. The GO in the prepared coating materials acts as a barrier to the permeate medium due to the labyrinth effect of the layered structure. Polymer materials also achieve water and energy depletion via intermolecular interactions among molecular chains (e.g., electrostatic interactions and hydrogen bonding) during which the presence of water cross-links the polymers through covalent bonds, triggering the self-healing properties of the polymers. The Ce (IV)/PEI-C/PAA-GO coating also showed an excellent self-healing effect due to the reversible dynamic interaction between the entangled molecular chains and the corrosive medium, and the corrosion resistance was found to be significantly improved.

## 2. Results and Discussions

### 2.1. Preparation and Characterization of Multilayer Coatings

Mg alloys are vulnerable to corrosive electrolytes; therefore, we placed a conversion layer based on cerium on top of the AZ31B substrate. The Mg was immersed for 30 min in a 0.05 M solution of cerium nitrate, followed by 30 min of sintering heating at 80 °C to oxidize some of the Ce(OH)_3_ ions into Ce(O)_2_ [32]. The obtained sample was identified as Ce(IV). The prefabricated natural (hydrogen oxide) membrane on the AZ31B substrate was instantly dissolved when it touched the pH 5.0 cerium conversion treatment solution. Immediately following the dissolving of the oxide (hydride) coating, the first anodic reaction occurred (dissolution of magnesium) [33]:Mg → Mg^2+^ + 2e^−^(1)
followed by spontaneous cathode reactions (proton reduction):2H_2_O + 2e^−^ → 2OH^−^ + H_2_(2)

At the interface between the substrate and solutions, the OH^−^ generated during the reaction caused a rise in pH. The following reactions occurred when the interface pH was high enough for Ce^3+^ to undergo hydrolysis and precipitate cerium hydroxide/oxide. (3)–(5) [34]:Ce^3+^ + 3H_2_O → Ce(OH)_3_ + 3H^+^(3)
and/or
Ce^3+^ + 2H_2_O → Ce(OH)_2_(4)
Ce(OH)_2_^2+^ → CeO_2_ + 2H^+^(5)

Polymer films made by the standard LbL-assembly processes often create an interpenetrating interface between adjacent layers due to polymer chain rearrangement, i.e., polymer “in-and-out” diffusion during deposition. This polymer diffusion is responsible for the rapid growth of LbL-assembled polyelectrolyte films. The PAA-GO polymer can be adsorbed on the surface of multilayer films with PEI-C as the outer layer due to supramolecular interactions (electrostatic interactions and hydrogen bonding) between PEI and PAA. Under neutral conditions, when the film is immersed in a PAA (pH 7.0) solution, it can not only diffuse into the film, but the carboxyl group in PAA can also create hydrogen and covalent bonds between the amino group in the PEI or the partially oxidized catechol. The deposition of multilayer films depends on the coexistence of supramolecular contacts and covalent bonds between adjacent layers, supporting the polymer with mobility and reversible interactions between molecular chains (The chemical synthesis of PEI-C and PAA-GO is shown in Figure 1c,d).

As seen in Figure 2a1, the cerium-based conversion layer completely covered the AZ31B substrate surface, and slight cracks were formed on the Ce(IV) sample. Hamlaoui et al. [35] suggested that these microcracks are related to the formation of bubbles on the surface of the Mg alloy during deposition and dehydration processes or the shear stress between the Mg alloy substrate and the deposited layer of the coating material [36]. These cracks encourage mechanical bonding of subsequent multilayer polymer deposition [23,37]. The EDS mapping patterns in Figure 2a2 show that the Ce element is evenly distributed on the substrate surface, proving the successful preparation of the Ce conversion film. Cerium-based conversion layers improved corrosion resistance and adhesion between the polymer coating and the Mg alloys substrate [32]. In order to compare the effects of the introduction of catechol and graphene oxide, samples with and without catechol and graphene oxide were prepared and identified as Ce (IV)/PEI-C/PAA-GO, Ce(IV)/PEI/PAA, and Ce(IV)/PEI-C/PAA multilayer coatings, respectively. In the case of the deposition of PEI-C and PAA-GO, Figure 2b–d shows that the surface of all the samples is crack-free and uniformly planar, whereas the surfaces of the samples with catechol become rougher and denser. The addition of C enables the formation of polyamines in the coating, producing an extremely rough surface morphology. Studies have shown that dopa molecules can react with different chemical groups, such as amines and thiols, to form covalent bonds. For example, the amine group in PEI reacts with the catechol group in dopamine to form a rich covalent interaction, resulting in a denser coating. In addition, PEI-C can be strongly adsorbed on the cerium-based conversion layer because the polyhydroxy structure of the catechol provides effective anchor points for the coating. Ce(IV)/PEI-C/PAA-GO has a rougher surface, a denser thin layer, and a more wrinkled morphology than the smooth surface of the graphene oxide-free samples. In addition, the EDS of Ce(IV)/PEI-C/PAA-GO shows the elemental composition of the dominance of carbon atoms, confirming the deposition of GO on the surface of the sample. GO has a certain uniformity in the coating distribution and fills the original major defects of the GO under the PAA wrapping.

Figure 3 depicts the TEM and HRTEM images of the Ce(IV)/PEI-C/PAA-GO supramolecular polymer coating. As shown in Figure 3a, the lamellar folded structure of GO flakes can be directly observed. The GO is uniformly dispersed throughout the polymer matrix, which is wrapped around the GO flakes, as seen in Figure 3b. We attribute this observation to the presence of PAA on the surface of the GO layers, which acts as a spacer between the GO sheets, preventing them from build-up and agglomeration. The lattice bands of these nanoparticles have a facet spacing of d = 0.22 nm, which can be attributed to the (111) plane of the CeO_2_ nanoparticles (Figure 3c); this observation agrees with that obtained from XRD experiments. The elemental diagram of the Ce (IV)/PEI-C/PAA-GO sample is overlaid by C, N, O, and Ce, as shown in Figure 3d. Thus, the uniform distribution of C also indicates the excellent dispersion of GO.

In the ^1^H NMR of PEI-C (Figure 4a), the peaks (a) at 6.63 ppm and 6.61 ppm and (b) at 6.59 ppm are characteristic peaks of the aromatic ring proton hydrogen, and no characteristic peak of the oxidized catechol (7.56~7.85 ppm) can be observed in the spectrum. It confirms that the catechol structure was successfully copolymerized with PEI.

X-ray diffraction (XRD) was used to characterize the composition of the coating. It can be noted that the relatively narrow peaks in Figure 4b belong to AZ31B and that the broader peak appearing at 2theta (28°) is attributed to diffraction peaks associated with cerium oxide. The broadening of this peak also confirms the presence of the CeO_2_ nanoparticles in the major superpositions. The XRD patterns of Ce(IV)/PEI/PAA, Ce(IV)/PEI-C/PAA, and Ce(IV)/PEI-C/PAA-GO are identical. The most likely explanation for these findings is that the amorphous PEI and PAA structures could not be detected by XRD.

FT-IR was used to study the modification of the polymer coatings; spectra are shown in Figure 4c. In the spectrum of Ce(IV)/PEI/PAA, peaks at 3447 cm^−1^, 1632 cm^−1^, 1401 cm^−1^, and 1043 cm^−1^ can be detected. For Ce(IV)/PEI-C/PAA and Ce(IV)/PEI-C/PAA-GO, the surface of catechol is rich in benzene rings and hydroxyl groups, and the hydroxyl stretching vibration peak appears at 3702 cm^−1^. The characteristic peaks at 2919 cm^−1^ and 2850 cm^−1^ correspond to the symmetric and asymmetric stretching modes of the group in -CH_2_, respectively. The stretching vibration of Ce(IV)/PEI-C/PAA-GO and the epoxy group C-O of GO is located at 1118 cm^−1^.

The UV-Vis absorption spectrum also confirms that the catechol structure was successfully grafted into the PEI, as seen in Figure 4d. The UV-Vis absorption spectra of Ce(IV)/PEI-C/PAA and Ce(IV)/PEI-C/PAA-GO show a strong UV absorption at a wavelength of λ = 280 nm, which indicates that the coatings have a high amount of catechol structures.

### 2.2. Characterization of the Corrosion Resistance

Electrochemical impedance spectroscopy (EIS) was utilized to analyze the corrosion resistance of several samples immersed in a 3.5 wt% NaCl solution for 0.5 to 72 h.

The EIS plots of the AZ31B alloy immersion for 2 h, 6 h, 12 h, 24 h, 48 h, and 72 h are presented in Figure 5a,b. From the Nyquist plots in Figure 5a, there are one low-frequency inductive loop and one high- and one mid-frequency capacitive loop. Magnesium in neutral aqueous solutions tends to produce a porous oxide or hydroxide layer, which does not entirely cover the surface [38]. The dissolution of metals in the pores and cracks in the films is connected with a medium-frequency inductive loop. However, MgO/hydroxide films on the alloy surface are related to a high-frequency inductive loop [39]. Low-frequency inductive loops were thought to be caused by corrosive species that had been adsorbed onto the substrate’s surface. When immersion time increases, the Nyquist plot shows a gradual decrease in the size of the capacitance loop associated with the resistance, indicating that as the immersion time increased, the pitting corrosion of the magnesium alloy substrate became more severe.

For bare AZ31B alloy, the Bode phase angle diagram displays the three time constants. The time constants’ maximum phase angles appear to be approximately 100 Hz during the first two hours of immersion. However, as immersion time proceeds, the maximum phase angle drastically decreases and moves to a lower frequency, reducing the substrate’s disintegration. This reduction is thought to be due to forming a MgO/hydroxide layer on the surface, which progressively gets thicker over time [38]. Furthermore, the maximum phase angle shifts from around 25° to 35°, indicating that the MgO/hydroxide layer is porous in nature. At the same time, flaws due to the increased development of free corrosion products cause an increase in layer thickness [40].

The impedance spectra in Figure 5e–h depict the composite coating’s evolution over 72 h while submerged in a NaCl solution. This composite coating evolved very differently from the bare coating. Mg alloys only exhibit one high-frequency capacitance ring, suggesting that corrosion happens on the alloy’s surface. Figure 5c,e,g show the Nyquist plot of composite-coated substrates, which shows the low-frequency capacitance ring. For all three coatings, the high-frequency capacitance ring is significantly bigger in diameter, and at least two orders of magnitude larger in impedance, than the bare AZ31B alloy. It is demonstrated that the coating with the multilayered film significantly enhances the corrosion resistance of the Mg alloy. The high-frequency capacitance ring grew larger as the soaking duration extended to 8 h, while the low-frequency coated capacitance ring vanished, the size of the capacitive ring of Ce(IV)/PEI-C/PAA-GO was the largest, and the impedance modulus at f = 0.1 Hz was 3.2 × 10^4^ Ω·cm^2^, higher than all other samples. After 48 h of soaking, the capacitive ring radius of the Ce(IV)/PEI/PAA coating decreased dramatically. The pattern of the curves was similar to that of the Mg substrate, and the |Z|(f = 0.1 Hz) of Ce(IV)/PEI/PAA was decreased to 1.2 × 10^3^ Ω·cm^2^. The presence of micropores and poor barrier properties will lead to protection failure, causing severe corrosion of samples. Comparably, Ce(IV)/PEI-C/PAA and Ce(IV)/PEI-C/PAA-GO with |Z|(f = 0.1 Hz) remained stable at 1.7 × 10^4^ Ω·cm^2^ and 3.4 × 10^4^ Ω·cm^2^, respectively, which was probably due to the introduction of catechol to make the Ce(IV)/PEI-C/PAA and Ce(IV)/PEI-C/PAA-GO supramolecular polymer bond better with the Mg alloy substrate. Ce(IV)/ PEI-C/PAA failed after 72 h of immersion and the |Z|(f = 0.1 Hz) decreased from a maximum (48 h) of 1.7 × 10^4^ Ω·cm^2^ to 1.1 × 10^4^ Ω·cm^2^. The capacitance ring radius of Ce(IV)/PEI-C/PAA-GO was: slightly reduced, the value of the |Z|(f = 0.1 Hz) of 2.9 × 10^4^ Ω·cm^2^, 1.5 orders of magnitude higher than Ce(IV)/PEI/PAA, and higher than all other samples. Ce(IV)/PEI-C/PAA-GO achieved the best electrochemical performance and long-term protection.

The positive peak in the Bode phase diagram represents the inductive response, and the negative peak represents the conductive response [41]. The positive peak in the low- and mid-frequency areas of the Ce(IV)/PEI/PAA Bode phase diagram migrated to the right with the increase in soaking time, and a negative peak and a positive peak in the low-frequency range (0.1 Hz–1 Hz) appeared as the length of protection was raised to 48 h. The inductive and conductive response of the coating indicates that the metal begins to dissolve, corrosion points begin to form, and the supramolecular coating is contaminated with corrosion (containing H_2_O and Cl^−^, etc.). As the coating deteriorates, the metal becomes more susceptible to attack from the corrosive medium, agreeing with the Nyquist findings and the Bode plot’s decreased |Z| value. The Ce(IV)/PEI-C/PAA layer also shows the same phase characteristics after immersion for 72 h. Nevertheless, GO’s barrier properties prevent further corrosion of the substrate. During the protection period, the GO-based coating exhibits superior corrosion resistance compared to the coating that does not contain GO.

The equivalent circuit diagram illustrated in Figure 6 is used to calculate the change in the coating’s resistance to determine how much the coating contributes to the protection process. Table 1 displays the resistance values for various times obtained using the equivalent circuit schematic. Rx, Cx, and Lx are symbols for the circuit’s resistance, conductor, and inductance, respectively. In addition, the fitted impedance parameters are exhibited in Table 1. When the frequency is equal to zero, the difference between the solution resistance and the actual impedance of the Nyquist plot is known as polarization resistance (Rp) [42]. Using this formulation as a guide, we can determine the equation for the equivalent Rp model as follows:(6)$$\frac{1}{R_{p}}=\frac{1}{R_{1}}+\frac{1}{R_{2}}+\frac{1}{R_{3}}$$
(7)$$\frac{1}{R_{p}}=\frac{1}{R_{1}+R_{2}+R_{4}}$$
(8)$$\frac{1}{R_{p}}=\frac{1}{R_{1}+R_{2}+R_{4}}+\frac{1}{R_{3}}$$

The R_P_ values of the samples containing GO were higher than those of the other samples (as shown in Figure 7b), demonstrating the significant GO contribution to corrosion resistance. For example, an Rp value of the Ce(IV)/PEI-C/PAA-GO sample increased to 3.5 × 10^4^ Ω·cm^2^ at 48 h. Even after 72 h, it was able to maintain its excellent performance. In contrast, for the other coatings, at different periods, the R_p_ values were reduced to minimal values, indicating that the coating was cracked or had even peeled off at this point. These results coincide with the EDS test of the supramolecular coating immersed in a NaCl solution for 72 h.

Figure 8 shows the polarization curves of samples soaked in a 3.5%wt NaCl solution for 72 h in electrochemical testing; the fitting results are shown in Table 2. The bare AZ31B alloy has a larger corrosion current density (Icorr) of 2.225 × 10^−6^ A/cm^2^ and a corrosion potential of −1.364 V. The Ecorr of the Ce(IV)/PEI/PAA coating is higher than that of the bare AZ31B alloy, accelerating the corrosion process. The premature destruction of Ce(IV)/PEI/PAA makes the multilayered film a reservoir layer for the corrosive electrolyte. The Icorr (0.942 × 10^−6^ A/cm^2^) value of the Ce(IV)/PEI-C/PAA-GO-coated Mg alloy is significantly lower. The anodic reaction in electrolytes is significantly inhibited by the protective coating, which has outstanding anti-corrosion properties.

The corrosion protection mechanism of the composite coating is further explored by the schematic diagram of different coating systems, as shown in Figure 9. For the simple PEI/PAA coating (Figure 9a), the corrosive elements are prone to penetrate inside the coating due to the tiny defects and cracks in the coating and the hydrophilic nature of the coating matrix. When electrolytes, water, dissolved oxygen, and chlorides reach the metal surface, a series of electrochemical and redox reactions occur at the interface between the coating and the metal surface, leading to the failure of the coating system. For the PEI-C/PAA coatings containing dopa molecules (Figure 9b), the entanglement density of PEI and PAA during growth is enhanced by the strong adhesion property of dopa molecules, resulting in a dense coating that can inhibit electrolyte penetration. However, electrolytes still penetrate directly into the coating, and adding the GO filler to PEI-C/PAA has two distinct advantages (Figure 9c). When the coating is intact, it improves the barrier performance of the coating by creating a “labyrinth effect” that stretches the penetration route of the corrosive medium through a random, non-cohesive distribution inside the coating. When the coating is damaged, corrosive substances like chloride ions permeate the damaged film, and water enters along with them. In this case, the GO sheets block the intrusion of corrosive media, and water excites a reversible dynamic interaction between the entangled molecular chains and the corrosive media, causing the polymer to cross-link through covalent bonds, triggering the self-healing property of the polymer. Therefore, the polymer heals the damaged region and prevents the entry of corrosive substances. As a result, the Ce(IV)/PEI-C/PAA-GO supramolecular polymer exhibits good corrosion protection properties.

Scanning electron microscopy (SEM) and elemental analysis were used to examine the morphology of the bare AZ31B alloy and the coating immersed in a 3.5 wt% NaCl solution for three days. Figure 10 shows that the Ce(IV)/PEI/PAA coating has many loose corrosion products, which means that severe corrosion has occurred. Most of the coating has peeled away due to the lack of adhesion capability when the layer broke after 72 h of soaking. The Ce(IV)/PEI-C/PAA and Ce(IV)/PEI-C/PAA-GO supramolecular polymers showed a low density of corrosion products, but the former exhibited dense and subtle corrosion cracks. The Ce(IV)/PEI-C/PAA-GO coating was highly protective, and the coating remained unchanged after 72 h of soaking. Compared with the other three coatings that dissolved and peeled off to varying degrees, Ce(IV)/PEI-C/PAA-GO remained firmly adsorbed on the substrate. This result could be attributed to the intrusion of corrosive ions by the GO sheets and the catechol structure that improved the adsorption of the coating and made the entire coating structure more holistic. The EDS mapping also shows that a larger number of Cl and Na elements are blocked on the upside of the coating to protect the substrate from erosion. The Mg elements represent the corrosion products of Mg-rich compounds and Mg(OH). Under the protection of the Ce(IV)/PEI-C/PAA-GO coating, the Mg elements have very little distribution, confirming the high corrosion resistance.

### 2.3. Characterization of Self-Healing Properties

Self-healing is the most critical characteristic of advanced anti-corrosion coatings. The ability of the Ce(IV)/PEI-C/PAA-GO multilayers to heal arises from reversible supramolecular interactions, including electrostatic and hydrogen bonding interactions between adjacent layers, as well as the mobility of the polymer chains [43,44]. Supramolecular interactions between adjacent layers may be partially disrupted when artificially cleaving the polymeric multilayer. When the coating is in contact with water, hydrogen bonding in the water enhances the dynamic changes between the polymers and supramolecular chain interactions, leading to the repair of the multilayer structures by re-establishing the polymer network at the fracture site, subsequently completing the self-repair processes.

The ability of Ce(IV)/PEI/PAA, Ce(IV)/PEI-C/PAA, and Ce(IV)/PEI-C/PAA-GO to self-heal was observed by an optical microscope. Scratches approximately 10 μm wide were cut on the surface using a scalpel, then a drop of water was placed onto the surface. Figure 11 shows the extent of repair between 10 and 20 min for each of the three coatings. Within 20 min, all three layers showed a strong healing ability. The coating containing the GO sheets left a slightly more noticeable scratch than the other samples; this behavior can be attributed to the build-up of GO affecting the dynamic motion of the intermolecular hydrogen bonds and polymer chains, which weakens the interaction between PEI and PAA. Figure 12 shows the records of the three-dimensional morphology of the Ce(IV)/PEI-C/PAA-GO self-healing surface at different times with an ultra-deep-field microscope. It can be seen that in the presence of water, the Ce(IV)/PEI-C/PAA-GO coating was completely repaired within 20 min.

## 3. Materials and Methods

### 3.1. Materials and Reagents

The polyethyleneimine PEI (Mw.ca = 750,000) in a 50% aqueous solution, 3-(3, 4-dihydroxyphenyl)propionic acid, cerium nitrate hexahydrate, and graphene oxide powder were purchased from the Aladdin Biochemical Technology, Shanghai, China Co. The polyacrylic acid PAA (Mw.ca = 450,000) and 1-(3-dimethylaminopropyl)-3-ethylcarbodiimide (EDC) were purchased from the Maclean Biochemical Technology, Shanghai, China Co. The NaOH, H (NO)_3_ and absolute ethanol were purchased from the Sinopharm Chemical Reagent, Shanghai, China. The AZ31B magnesium alloy substrates is provided by Huatai Metal Materials, Guangdong, China Co. It is ground with 1200 grit SiC sandpaper before use and cleaned with ultrasonication with ethanol for 5 min. All reagents were analytical grade, and no further purification was necessary.

### 3.2. Synthesis of PEI-C and PAA-GO

A 1 g sample of PEI was dissolved in 100 mL of PBS solution and adjusted to a 5.5 pH with a solution of 1 M H (NO)_3_. Subsequently, 0.5067 g of (3,4-dihydroxy phenyl) propionic acid and 0.9033 g of EDC were added to the solution. The pH concentration in the reaction solution was maintained at 5.5, and the solution was stirred continuously at room temperature for 2 h to yield the PEI-C solution. For the PAA-GO solution, 0.3 g of PAA was dissolved in 100 mL of deionized water by stirring for 2 h at room temperature to fully dissolve the PAA. Following this, 0.01 g of graphene oxide powder was inserted into the PAA solution and dispersed ultrasonically for 30 min, then adjusted to a pH of 7 with a 1 M NaOH solution to yield a PAA-GO solution.

### 3.3. Preparation of Multilayer Coatings

Cerium nitrate (2.17 g) was dissolved in 100 mL of deionized water to make a solution of 0.05 M cerium nitrate hexahydrate. The AZ31B magnesium plate was dipped in a solution of cerium nitrate hexahydrate at room temperature (approximately 23 °C) for 30 min and then heated to 80 °C for 30 min to evaporate excess water; this sample was designated as Ce(IV). The sample Ce(IV) was then deposited in a PEI-C solution and a PAA-GO solution 10 times in alternation. The soak time for both PEI-C and PAA-GO was 5 min, and, during each deposition step, the samples were rinsed in deionized water for 2 min.

### 3.4. Characterizations

Scanning electron microscopy (JSM-IT500, JEOL, Japan) was used to characterize the surface morphology of the material, EDS spectra were obtained at 9.0 kV. Transmission electron microscopy (JEM 2100, JEOL, Tokyo, Japan) micrographs is characterized under the abeam acceleration of 200 KV to observe the distribution of graphene oxide nanosheets and CeO_2_ nanoparticles. An X-ray diffractometer (D/max2500 vpc, Rigaku, Japan) was utilized for the structural analysis. Fourier infrared spectroscopy (Spectrum 100, PerkinElmer, Waltham, MA, U.S.A) was used to study the chemical structure and bonding properties of the prepared materials. Structural changes were corroborated using UV-Vis (Cary 5000, Agilent, Santa Clara, CA, U.S.A) spectrophotometry. Analysis of the polymeric materials was by ^1^H NMR analysis was performed using a spectrometer (AV600, Bruker, Germany). Electrochemical testing was conducted via an electrochemical power plant (Chenhua Instruments, CHI 660e, Shanghai, China). Electrochemical impedance spectroscopy (EIS) and Tafel assays were carried out using a platinum electrode (counter electrode), standard glyceryl electrodes (reference electrode), magnesium substrates (working electrode), and a 3.5 wt% NaCl solution to characterize the corrosion resistance of the coating; tests were performed at a frequency of 0.1 Hz~100 kHz. The initial potential was set equal to the open-circuit potential (OCP). The Tafel measurements were taken between −2 V and 1 V of OCP. The self-healing process of the materials was observed and recorded with an optical microscope (DMI 3000M, LAS V4.3, Leica, Nussloch, Germany) and an ultra-deep-field microscope (VHX-2000, KEYENCE, Shanghai, China).

## 4. Conclusions

A novel self-healing anti-corrosion supramolecular polymer coating was successfully prepared on the surface of a Mg alloy. The cerium-based conversion layer inhibited the corrosion of the AZ31B alloy by GO and catechol, which enhanced the adhesion between the supramolecular polymer coating and the substrate and made the coating dense. Layer-by-layer (LbL) assembly of PEI-C and PAA promoted intermolecular interactions and endowed the self-healing properties. The experimental results show that a 10 μm scratch healed completely within 20 min in the presence of water. The GO sheets endowed the supramolecular polymer coating with an excellent anti-corrosion performance by preventing contact between the corrosion medium and the substrate. The electrochemical behavior of supramolecular polymer coating was investigated by EIS; the impedance modulus of the Ce (IV)/PEI-C/PAA-GO coating was up to 3.2 × 10^4^ Ω·cm^2^, which is two orders of magnitude higher than that of the substrate. After soaking in a 3.5 wt% NaCl solution for 72 h, the corrosion current was 0.942 × 10^−6^ A/cm^2^, which was superior to the results for the other coatings in this work. The Ce(IV)/PEI-C/PAA-GO supramolecular polymer coating provides a new idea for developing multifunctional anti-corrosion coatings.

## Figures and Tables

**Figure 1 ijms-24-04981-f001:**
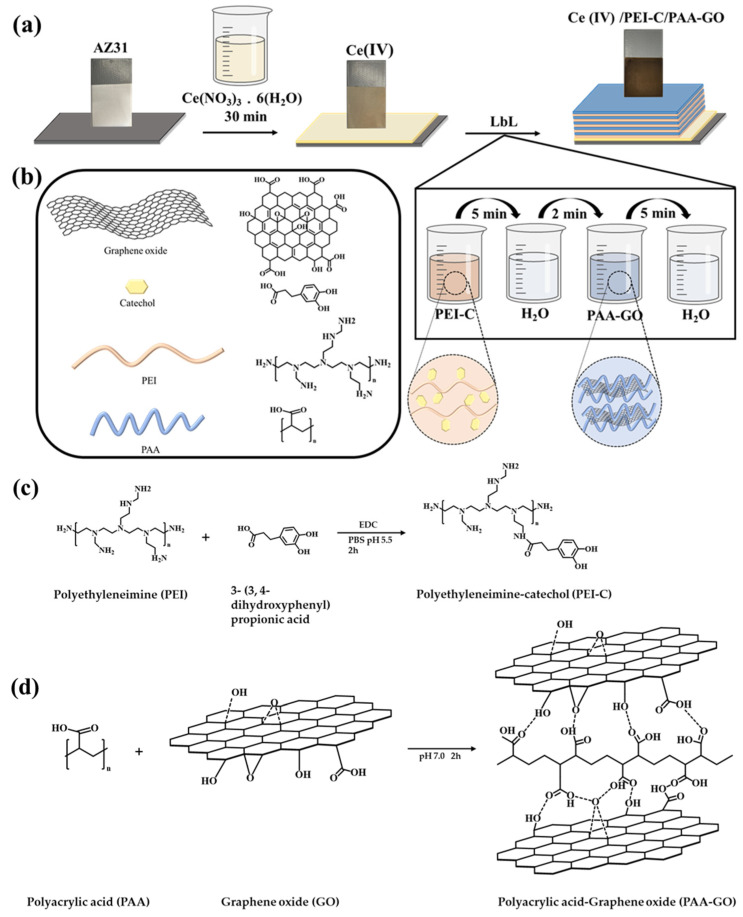
(**a**) The preparation process of the self-healing anti-corrosion coating and (**b**) the chemical structures of graphene oxide, catechol, PEI, and PAA. The chemical cross-linking scheme of (**c**) the polymer PEI with catechol-containing supramolecules and (**d**) the polymer PAA with graphene oxide.

**Figure 2 ijms-24-04981-f002:**
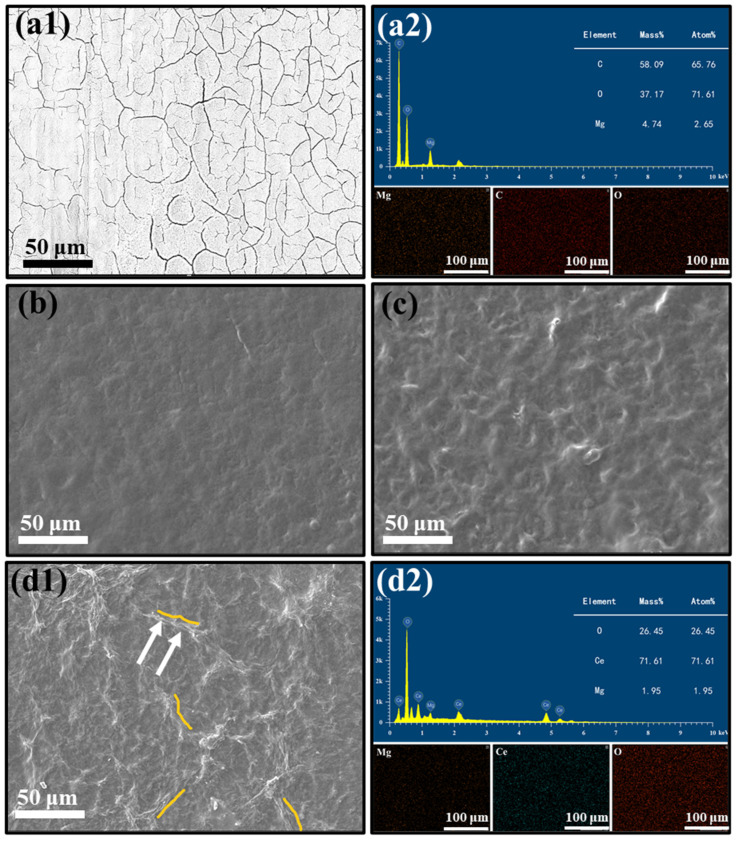
SEM images of the samples under different coatings: (**a1**) surface morphology of Ce(IV), (**a2**) EDS pattern of Ce(IV), (**b**) surface morphology of Ce(IV)/PEI/PAA, (**c**) Ce(IV)/PEI-C/PAA, (**d1**) Ce(IV)/PEI-C/PAA-GO, and (**d2**) EDS pattern of Ce(IV)/PEI-C/PAA-GO coating.

**Figure 3 ijms-24-04981-f003:**
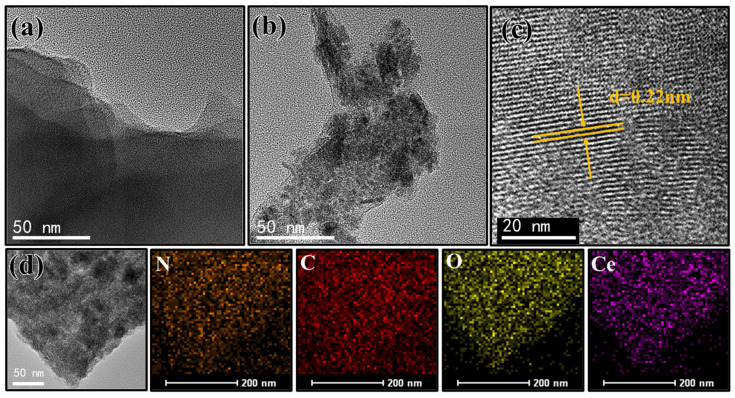
(**a**,**b**) TEM images of (**c**) HRTEM images and (**d**) energy dispersive x-ray energy spectrum mapping of Ce(IV)/PEI-C/PAA-GO.

**Figure 4 ijms-24-04981-f004:**
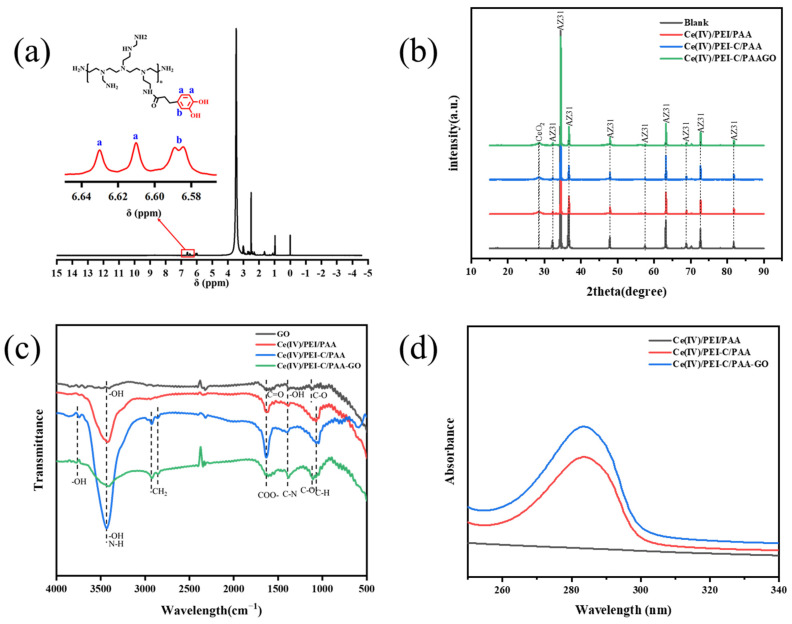
The compositional characterization of the different patterns of: (**a**) ^1^H NMR of PEI-C, (**b**) XRD, (**c**) FT-IR, and (**d**) UV-Vis.

**Figure 5 ijms-24-04981-f005:**
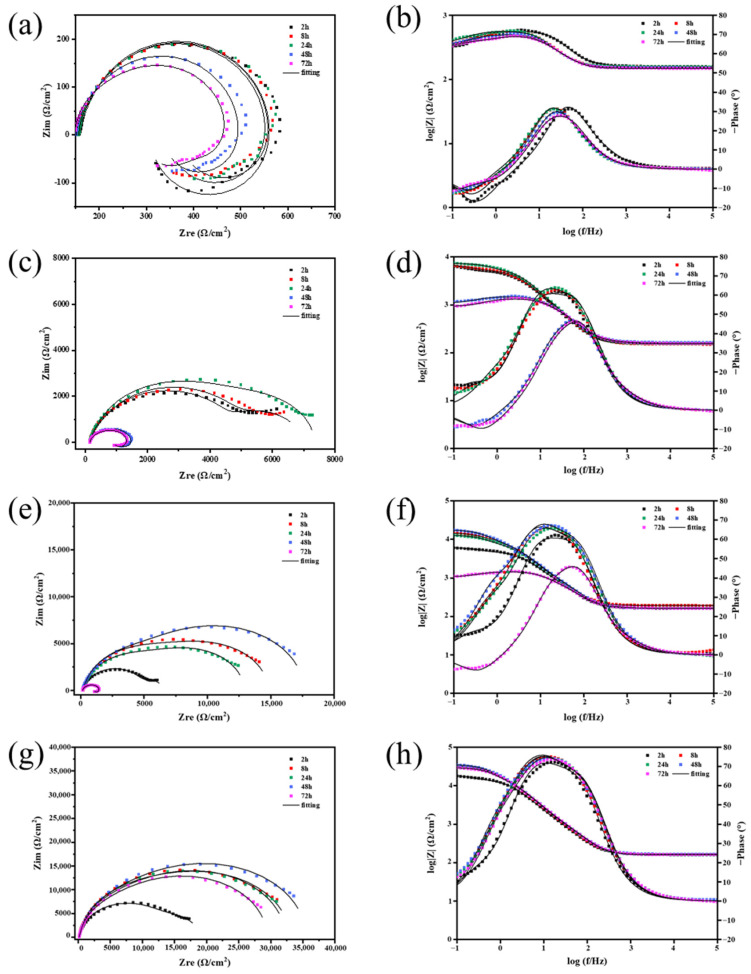
EIS of a series of immersion times of (**a**,**b**) Bare AZ31B alloy, (**c**,**d**) Ce(IV)/PEI/PAA, (**e**,**f**) Ce(IV)/PEI-C/PAA, and (**g**,**h**) Ce(IV)/PEI-C/PAA-GO in a 3.5 wt% NaCl solution.

**Figure 6 ijms-24-04981-f006:**
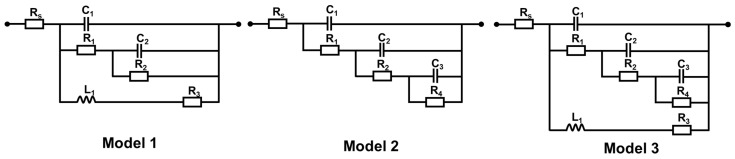
Three equivalent circuit diagrams.

**Figure 7 ijms-24-04981-f007:**
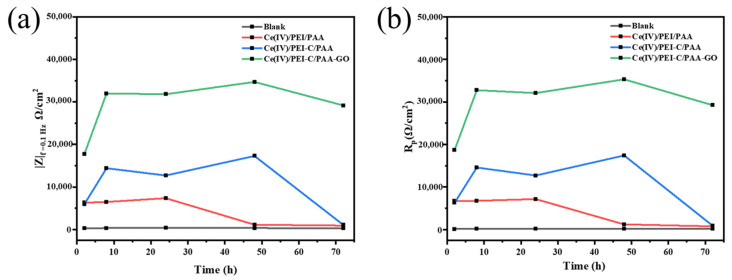
The variation of: (**a**) the impedance modulus (f = 0.1 Hz) and (**b**) polarization resistance with the immersion times for coated and bare AZ31B alloys of different compositions in a 3.5 wt% NaCl solution.

**Figure 8 ijms-24-04981-f008:**
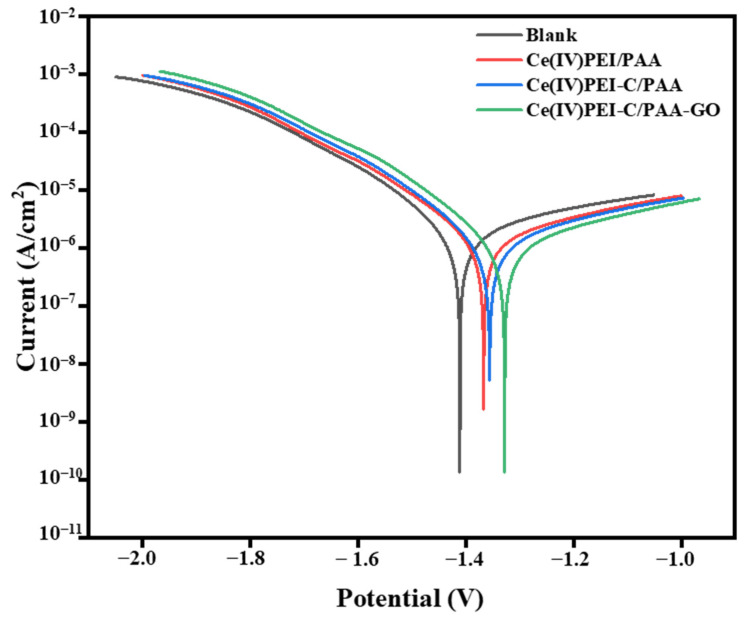
Polarization curves of bare AZ31B alloy and the alloy coated with different sample compositions immersed in a 3.5 wt% NaCl solution for 72 h.

**Figure 9 ijms-24-04981-f009:**
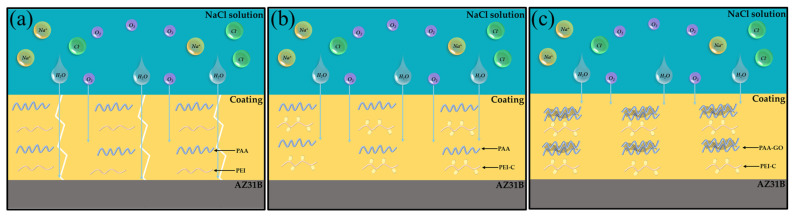
Corrosion mechanism diagrams of as-prepared coating systems for (**a**) Ce(IV)/PEI/PAA, (**b**) Ce(IV)/PEI-C/PAA, and (**c**) Ce(IV)/PEI-C/PAA-GO.

**Figure 10 ijms-24-04981-f010:**
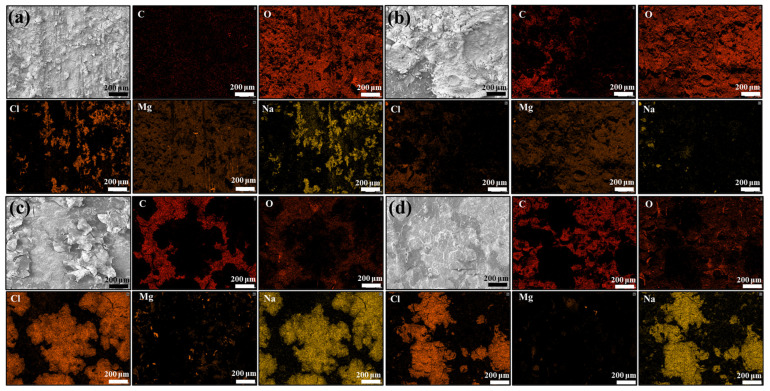
The microscopic morphology and EDS patterns of (**a**) bare AZ31B alloy, (**b**) Ce(IV)/PEI/PAA, (**c**) Ce(IV)/PEI-C/PAA, and (**d**) Ce(IV)/PEI-C/PAA-GO immersed in a 3.5 wt % NaCl solution for 72 h.

**Figure 11 ijms-24-04981-f011:**
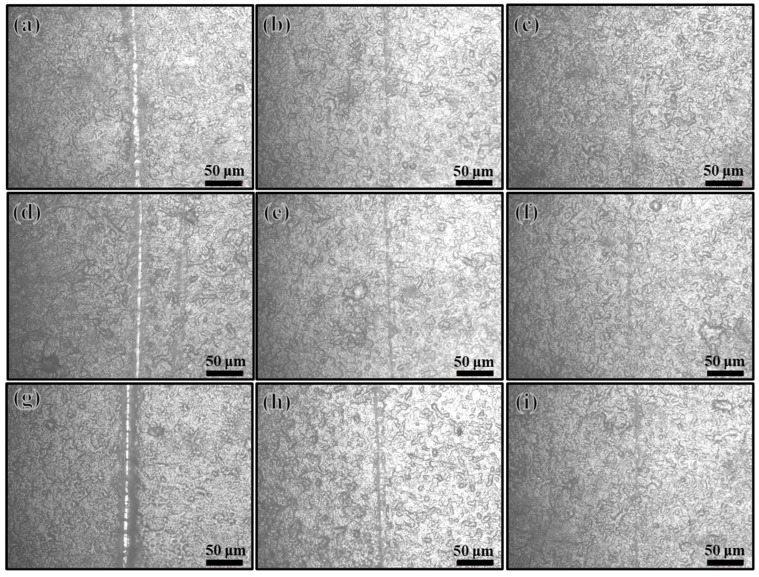
Self-healing tests of samples with different coatings in the presence of water: (**a**) Ce(IV)/PEI/PAA surface incision, (**b**,**c**) surface morphology of Ce(IV)/PEI/PAA self-healing in the presence of water for 10 and 20 min, (**d**) Ce(IV)/PEI/PAA surface incision, (**e**,**f**) surface morphology of Ce(IV)/PEI-C/PAA self-healing in the presence of water for 10 and 20 min, (**g**) Ce(IV)/PEI/PAA surface incision, (**h**,**i**) surface morphology of Ce(IV)/PEI-C/PAA-GO self-healing in the presence of water for 10 and 20 min.

**Figure 12 ijms-24-04981-f012:**
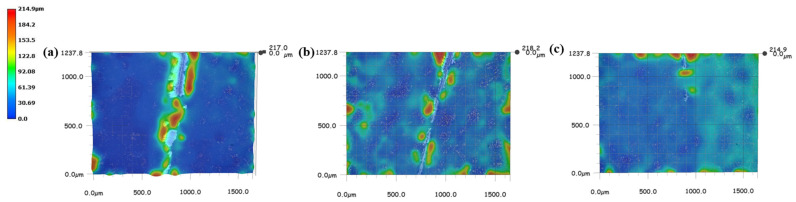
The 3D morphology of a Ce(IV)/PEI-C/PAA-GO-coated surface: (**a**) surface incision, (**b**) 10-min self-repair, and (**c**) 20-min self-repair.

**Table 1 ijms-24-04981-t001:** Electrical parameters for different coating systems immersed in a 3.5% NaCl solution; the values were obtained by fitting the EIS data.

Sample	Time (h)	R_s_ (Ω·cm^2^)	C_1_ (F/cm^2^)	R_1_ (Ω·cm^2^)	C_2_ (F/cm^2^)	R_2_ (Ω·cm^2^)	C_3_ (F/cm^2^)	R_3_ (Ω·cm^2^)	L_1_ (H·cm^2^)	R_4_ (Ω·cm^2^)	R_p_ (Ω·cm^2^)	Model
AZ31	2	159.3	1.36 × 10^−5^	303.7	4.22 × 10^−5^	104		237.4	211.3		150	1
	8	157.3	2.69 × 10^−5^	249	4.20 × 10^−5^	151.2		358.9	333.4		189	1
	24	157.2	2.74 × 10^−5^	235.9	3.74 × 10^−5^	169.2		460.8	528.6		215.6	1
	48	151.8	2.62 × 10^−5^	222	4.48 × 10^−5^	124.4		392	399		183	1
	72	148.7	2.47 × 10^−5^	194.4	4.38 × 10^−5^	118		388	343.6		173.1	1
Ce/PEI/PAA	2	154.3	7.10 × 10^−6^	1707	7.31 × 10^−6^	3034	4.43 × 10^−4^			1996	6737	2
	8	152.3	6.68 × 10^−6^	1923	7.53 × 10^−6^	3169	4.14 × 10^−4^			1676	6768	2
	24	152.3	6.68 × 10^−6^	1665	4.88 × 10^−6^	3781	1.29 × 10^−4^			1728	7174	2
	48	162.8	6.68 × 10^−6^	123.6	2.27 × 10^−6^	796.4	2.16 × 10^−5^	3195	1906	398.5	1236	3
	72	155.7	6.68 × 10^−6^	108	2.43 × 10^−6^	744.9	2.42 × 10^−5^	2213	935.2	336.9	773.8	3
Ce/PEI-C/PAA	2	163.4	6.68 × 10^−6^	1764	6.80 × 10^−6^	2990	3.86 × 10^−5^			1231	6330	2
	8	194.2	5.96 × 10^−6^	2414	4.19 × 10^−6^	7284	6.54 × 10^−5^			4884	14,582	2
	24	166.3	5.73 × 10^−6^	2013	4.32 × 10^−6^	6088	6.04 × 10^−5^			4607	12,708	2
	48	163.8	5.36 × 10^−6^	2042	3.61 × 10^−6^	7568	3.28 × 10^−5^			7810	17,420	2
	72	161.7	4.25 × 10^−6^	94.57	3.09 × 10^−6^	785.6	2.23 × 10^−5^	3018	1609	427.5	912.4	3
Ce/PEI-C/PAA-GO	2	165.4	5.46 × 10^−6^	3562	2.78 × 10^−6^	10,930	1.70 × 10^−4^			4234	18,726	2
	8	163.3	4.84 × 10^−6^	3567	1.94 × 10^−6^	20,180	3.26 × 10^−5^			9009	32,756	2
	24	159.6	4.43 × 10^−6^	3337	1.88 × 10^−6^	18,150	1.99 × 10^−5^			10,590	32,077	2
	48	159.6	4.42 × 10^−6^	3504	1.87 × 10^−6^	19,540	1.80 × 10^−5^			12,260	35,304	2
	72	159.6	4.41 × 10^−6^	3073	1.88 × 10^−6^	15,910	1.74 × 10^−5^			10,250	29,233	2

**Table 2 ijms-24-04981-t002:** The sample fitting results of the polarization curves.

Sample	E_corr_ (V)	I_corr_ (A/cm^2^)
Blank	−1.411	1.411 × 10^−6^
Ce(IV)/PEI/PAA	−1.366	1.401 × 10^−6^
Ce(IV)/PEI-C/PAA	−1.357	1.011 × 10^−6^
Ce(IV)/PEI-C/PAA-GO	−1.328	0.942 × 10^−6^

## Data Availability

Not applicable.

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
