# Peer review of "Preparation of a Mussel-Inspired Supramolecular Polymer Coating Containing Graphene Oxide on Magnesium Alloys with Anti-Corrosion and Self-Healing Properties"

_ijms, 2023, doi:10.3390/ijms24054981_

Round 1
Reviewer 1 Report
An interesting paper about the use of self-healing nanomaterial based coating to improve corrosion resistance in metallic surfaces. I have a couple of issues with the paper. These include:
1) The English needs to be checked, especially the consistent use of one tense and the passive voice. I noted the following examples
Line 117: Change "layer is completely" to "layer was completely"
Line 118: Change "slight cracks was formed" "slight cracks were formed"
Please revise the English in the rest of the text
2) Make sure that you are referring to the correct instrument in the text. On page 389 you refer to "light electron microscope" to image self healing while in materials and methods you refer to "light microscope" please correct which one was correct.
3) The results of the EIS examinations should be more in context. Can you include a table of other experimental results to show how this material compares?
4) The Microscope images are too small and low res to see properly. Please resize them. Please also make sure that the scale-bar is more visible.
5) Do you have more characterization details of the Graphene Oxide Powder used in these experiments? What is the average sheet size, degree of oxidation?
Author Response
请参阅附件

Reviewer 2 Report
The article presented by Zhang et al. and entitled: "Mussel-inspired supramolecular polymer coatings containing graphene oxide with anti-corrosion and self-healing properties" represents an interesting study on supramolecular polymer-composite coatings which exhibit anti-corrosive and self-healing properties and can be applied to Magnesium based alloys. I believe this work is of high quality, fits the scope of the journal and can be published after Major revision following the consideration of the below comments:
Comments:
(1) Abstract: Line 10: (all abbreviations such as PEI, PAA, GO) must be clearly defined. Also in Line 8: "we report".
(2) English in general should be carefully revised. Many sentences are confusing and are hard to follow.
(3) Line 32: Traditional coatings (Cite: Macromolecules 2010, 43, 23, 9598-9600. DOI: 10.1021/ma1019889)
(4) Line 69: "which" facilitates, Line 72: "was" grafted, Line 71: Coating (remove s)
Reviewer 3 Report
The current article “Mussel-inspired supramolecular polymer coatings containing graphene oxide with anti-corrosion and self-healing properties’ This article can be accepted after major corrections.
In abstract section must be mention full form of PEI-C/PAA
Experimental section according to journal format
During the deposition of CeO layer some cracks. How can remove/fill the cracks portion of CeO layer explain.
Fig 6 Model should be clear visible representation
Line no.211 when time increases loop size increases but figure is different shown.
Table 1 Rs values in CePEI/PAA at 48oC increase than decrease reason explains
Phase angle should be negative
Table 2 Ecorr value not in trends. Please explain the reason in details
Reviewer 4 Report
Zhang et al. examine the Mussel-inspired supramolecular polymer coatings containing graphene oxide with anti-corrosion and self-healing properties. The article is written nicely, explaining all the aspects related to the study. It will help researchers who are working in the same field. I recommend its acceptance after a major revision followed by the editorial correction.
1. Title should be modified as it does not give the scientific meaning of the research.
2. Abstract should contain key findings of the work, please add it.
3. There should be a space (gap) between the numerical values and the units.
4. Figure 3 is not properly visible, please update it.
5. The conclusion can be improved.
6. Please exchange old papers with newly published papers.
7. Please follow one homogeneous format style of referencing. Different styles of referencing can be seen.
8. Please check the entire manuscript and remove grammatical errors.
9. Most importantly plagiarism is around 30% in this current version of the manuscript, please reduce it by <20%.
10. Compared to other published materials, what does it bring to the field?
11. What makes this work novel?
12. The author should contrast this recent study with previously published authoritative data.
13. Authors should clarify whether their findings are in line with the facts and arguments made and whether they answer the primary query.
14. The social and economic effects of corrosion might be added by the authors.
Round 2
Reviewer 2 Report
The authors have satisfactory addressed the comments. The manuscript can be accepted in the present form.
Author Response
We appreciate your time.
Reviewer 3 Report
Figure 5 Bode plots x and y axis actual values not in log, just author written in legend log. author must be values converted in log, now phase angle values is OK.
Reviewer 4 Report
The authors have resolved all the issues raised by the reviewer. Hence, I recommend its acceptance.
Author Response
We appreciate your time.